# Admission Serum Potassium Levels in Hospitalized Patients and One-Year Mortality

**DOI:** 10.3390/medicines7010002

**Published:** 2019-12-30

**Authors:** Charat Thongprayoon, Wisit Cheungpasitporn, Panupong Hansrivijit, Michael A. Mao, Juan Medaura, Tarun Bathini, Api Chewcharat, Stephen B. Erickson

**Affiliations:** 1Division of Nephrology and Hypertension, Department of Medicine, Mayo Clinic, Rochester, MN 55905, USA; charat.thongprayoon@gmail.com (C.T.); api.che@hotmail.com (A.C.); Erickson.Stephen@mayo.edu (S.B.E.); 2Division of Nephrology, Department of Internal Medicine, University of Mississippi Medical Center, Jackson, MS 39216, USA; jmedaura@umc.edu; 3Department of Internal Medicine, University of Pittsburgh Medical Center Pinnacle, Harrisburg, PA 17101, USA; p.hansrivijit@gmail.com; 4Division of Nephrology and Hypertension, Mayo Clinic, Jacksonville, FL 32224, USA; mao.michael@mayo.edu; 5Department of Internal Medicine, University of Arizona, Tucson, AZ 85721, USA; tarunjacobb@gmail.com

**Keywords:** potassium, electrolytes, hypokalemia, hyperkalemia, hospitalization, mortality

## Abstract

**Background:** The aim of this study was to assess the relationship between admission serum potassium and one-year mortality in all adult hospitalized patients. **Methods:** All adult hospitalized patients who had an admission serum potassium level between the years 2011 and 2013 at a tertiary referral hospital were enrolled. End-stage kidney disease patients were excluded. Admission serum potassium was categorized into levels of ≤2.9, 3.0–3.4, 3.5–3.9, 4.0–4.4, 4.5–4.9, 5.0–5.4, and ≥5.5 mEq/L. Cox proportional hazard analysis was performed to assess the independent association between admission serum potassium and one-year mortality after hospital admission, using an admission potassium level of 4.0–4.4 mEq/L as the reference group. ** Results:** A total of 73,983 patients with mean admission potassium of 4.2 ± 0.5 mEq/L were studied. Of these, 12.6% died within a year after hospital admission, with the lowest one-year mortality associated with an admission serum potassium of 4.0–4.4 mEq/L. After adjustment for age, sex, race, estimated glomerular filtration rate (eGFR), principal diagnosis, comorbidities, medications, acute kidney injury, mechanical ventilation, and other electrolytes at hospital admission, both a low admission serum potassium ≤3.9 mEq/L and elevated admission potassium ≥5.0 mEq/L were significantly associated with an increased risk of one-year mortality, when compared with an admission serum potassium of 4.0–4.4 mEq/L. Subgroup analysis of chronic kidney disease and cardiovascular disease patients showed similar results. **Conclusion:** This study demonstrated that hypokalemia ≤3.9 mEq/L and hyperkalemia ≥5.0 mEq/L at the time of hospital admission were associated with higher one-year mortality.

## 1. Introduction 

Abnormalities in potassium balance are common medical problems encountered among hospitalized patients [1,2,3]. The prevalence of serum potassium abnormalities (hypo- and hyperkalemia) in hospitalized patients is reported to be as high as 48% [4,5,6]. Studies have demonstrated significant impacts of both hypokalemia and hyperkalemia on major adverse cardiac events (MACEs) and increased overall mortality in hypertensive, dialysis, and congestive heart failure (CHF) patients [1,7].

Hypokalemia is known to increase the risk of supraventricular and ventricular arrhythmias, especially when serum potassium is <3.5 mEq/L [7,8,9,10]. A large cohort of 2.6 million patients found that hypokalemia was significantly associated with a 2-fold increased risk of in-hospital mortality among patients with a diagnosis of CHF, regardless of causes of hypokalemia [11]. In addition, among patients with chronic kidney disease (CKD), hypokalemia is associated with higher all-cause mortality [2,12]. While hyperkalemia is uncommon (<5%) in the general population [13], it may affect up to 10% of all hospitalized patients, especially in those with CKD, diabetes mellitus (DM), cardiovascular disease (CVD), or those taking renin-angiotensin-aldosterone system (RAAS) inhibitors [2,6,13,14,15]. Studies have consistently demonstrated increased in-hospital mortality among patients with hyperkalemia, especially in those with CKD and CVD diagnoses [6,16,17].

Recently, we conducted a large retrospective cohort study, evaluating the association between admission serum potassium and in-hospital mortality [6] among all hospitalized patients. We demonstrated that an admission serum potassium <4.0 mEq/L and >5.0 mEq/L were associated with higher in-hospital mortality [6]. However, data on the impact of admission potassium levels on long-term survival beyond hospital stay remains unclear. Thus, we conducted this study to evaluate the impact of admission serum potassium levels on one-year mortality in hospitalized patients.

## 2. Methods

### 2.1. Patient Population

All adult patients admitted to Mayo Clinic Rochester, Minnesota, USA, from 1 January 2011 to 31 December 2013, were included. Patients who had no serum potassium measurement within 24 h of hospital admission or patients who were on chronic dialysis prior to admission were excluded. Figure 1 demonstrates the flow chart of the study. Only the first hospital admission was included for patients with recurring hospital admissions during the study period. This study was reviewed and approved by the Mayo Clinic institutional review board (IRB number: 15-000024; Approval Date: 2 April 2015). The informed consent was waived due to the minimal risk nature of the study, but all included patients provided research authorization for their patient data use.

Clinical characteristics, demographic information, and laboratory data were collected from the institutional electronic medical record system. The admission serum potassium level was defined as the first serum potassium level within 24 h of hospital admission. Serum potassium was measured using the ion-selective electrode (indirect potentiometry) method. Estimated glomerular filtration rate (eGFR) was calculated based on age, sex, race, and admission serum creatinine, using the chronic kidney disease epidemiology collaboration equation [18]. Chronic kidney disease was defined as eGFR of less than 60 mL/min/1.73 m^2^. The Charlson comorbidity score [19] was computed to assess for co-morbidity burden at the time of admission. Principal diagnoses were grouped based on ICD-9 codes at admission. The primary outcome was one-year mortality after hospital admission. A patient’s vital status was obtained from our institutional registry and the social security death index database.

### 2.2. Statistical Analysis

Continuous variables were summarized as mean ± standard deviation (SD), and categorical variables as a number with percentage, respectively. The difference in baseline demographics and clinical characteristics were assessed between the admission potassium groups using ANOVA for continuous variables and the Chi-square test for categorical variables. Admission serum potassium was categorized into 7 groups in order to assess for the non-linear association: ≤2.9, 3.0–3.4, 3.5–3.9, 4.0–4.4, 4.5–4.9, 5.0–5.4, and ≥5.5 mEq/L. Admission potassium of 4.0–4.5 mEq/L was selected as the reference group for outcome comparison because this range was within the referenced normal range and associated with the nadir for one-year mortality. Patient survival was measured from the initial hospital admission to death or the last inpatient/outpatient follow-up visit. One-year mortality risk was estimated using the Kaplan–Meier plot, and it was compared between admission serum potassium groups using the log-rank test. Multivariable cox proportional hazard analysis was constructed to assess the independent risk of one-year mortality based on admission serum potassium with adjustment for priori-defined covariates. Hazard ratio (HR) with 95% confidence interval (CI) was reported and adjusted for age, sex, race, eGFR, principal diagnosis, Charlson comorbidity score, comorbidities, medications, acute kidney injury, mechanical ventilation, and other electrolytes at hospital admission. Subgroup analysis of hospitalized patients with an admission principal diagnosis of CKD and cardiovascular disease was performed. A two-tailed *p*-value of less than 0.05 was considered statistically significant. All analyses were performed using JMP statistical software (version 10, SAS Institute, Cary, NC, USA, 2012).

## 3. Results

### 3.1. Baseline Characteristics

A total of 73,983 hospitalized patients were included in this study. The mean age was 61 ± 18 years, and 53% were male. The mean eGFR was 80 ± 26 mL/min/1.73 m^2^. A total of 15,716 (21%) patients were primarily admitted for cardiovascular disease. The mean admission serum potassium was 4.2 ± 0.5 mEq/L. An admission serum potassium of ≤2.9, 3.0–3.4, 3.5–3.9, 4.0–4.4, 4.5–4.9, 5.0–5.4, and ≥5.5 mEq/L was seen in 0.9%, 5%, 23%, 40%, 22%, 6%, and 2% of patients, respectively. Table 1 shows the clinical characteristics based on the admission serum potassium.

### 3.2. Admission Serum Potassium and One-Year Mortality

Of 73,983 patients, 12.6% died within one year after hospital admission. Kaplan–Meier plot estimated one-year mortality at 22.6% in admission serum potassium group of ≤2.9, 16.4% in 3.0–3.4, 12.1% in 3.5–3.9, 10.6% in 4.0–4.4, 12.4% in 4.5–4.9, 18.6% in 5.0–5.4, and 25.8% in ≥5.5 mEq/L (Figure 2).

In adjusted analysis, a low admission serum potassium of ≤2.9, 3.0–3.4, and 3.5–3.9 mEq/L was significantly associated with an increased risk of one-year mortality with HR of 1.67 (95% CI 1.39–2.01), 1.36 (95% CI 1.23–1.49), and 1.19 (95% CI 1.11–1.26), respectively. Elevated admission potassium of 5.0–5.4 and ≥5.5 mEq/L were also significantly associated with an increased risk of one-year mortality with HR of 1.30 (95% CI 1.19–1.41) and 1.62 (95% CI 1.43–1.83), respectively. There was no difference in one-year mortality when admission serum potassium ranged from 4.0–4.9 mEq/L (Table 2).

Subgroup analysis of patients admitted with a principal diagnosis of CKD and CVD showed similar associations between admission serum potassium and risk of one-year mortality (Table 3 and Table 4). Of note, in the subgroups of CKD and CVD patients, the increased risk of one-year mortality associated with low admission serum potassium ≤2.9 mEq/L exceeded the risk associated with elevated admission serum potassium ≥5.5 mEq/L.

## 4. Discussion

Regardless of the principal diagnosis, both hypo- and hyperkalemia were associated with an increased one-year mortality rate when compared to the normokalemia group. We demonstrated that a serum potassium level <4.0 mEq/L or >5.0 mEq/L was associated with an increased risk of death within one year of hospitalization. This mortality risk increased in accordance with the severity of hypo- and hyperkalemia.

The risk was highest in patients with serum potassium level ≤2.9 mEq/L and ≥5.5 mEq/L (HR 1.67 and 1.62, respectively). Our findings were consistent with those previously described by Collins et al. [2]. In their study, they reviewed 911,698 patients mainly in the outpatient setting and discovered a continuous U-shaped relationship between dyskalemia and all-cause mortality in the entire population, as well as in the cohorts with CHF, CKD, and diabetes. The median follow-up time was 18 months. In line with our research, we suggested that the optimal potassium level among hospitalized patients should be kept between 4.0–5.0 mEq/L to prevent cardiovascular adverse outcomes.

Our study also suggested that hypokalemia might be associated with a greater risk of death when compared to hyperkalemia in patients with a principal admission diagnosis of cardiovascular disease or GFR <60 mL/min/1.73 m^2^. From our multivariate analysis of these subgroups of patients, the trend for increased mortality was observed when serum potassium level was <4.0 mEq/L and when serum potassium level was ≥5.5 mEq/L (not ≥5.0 mEq/L like primary analysis). However, this finding might be underpowered as hyperkalemia is rare in the general population, making available sample size for subgroup analysis low. Nonetheless, our result suggested that hypokalemia in CKD and cardiovascular patients was alarming and warranted medical attention.

The mechanism by which dyskalemia is associated with increased mortality can be hypothesized due to the effect of potassium ions on cardiac myocytes. Hypokalemia promotes triggered arrhythmias by reducing cardiac repolarization reserve and increasing intracellular Ca^2+^ in cardiac myocytes [20]. This effect paradoxically increases the excitability of cardiac myocytes, predisposing to ventricular arrhythmias. Other mechanisms have been proposed in animal or in vitro models [21]. In contrast, systemic hyperkalemia enhances repolarization reserve by increasing K^+^ channel conductance, inducing post-repolarization refractoriness as manifested by tall peaked T-waves. Widening of the QRS complex in hyperkalemia is caused by the induced depolarization of the resting membrane potential [20]. This pathophysiology and its impact can be seen in clinical practice. For instance, hyperkalemia following coronary occlusion results in more arrhythmogenicity and fatal outcomes. Following coronary occlusion, interstitial K^+^ concentration rises rapidly in the central ischemic zone, resulting in depolarization of injury currents. This current can re-excite non-ischemic tissue to induce re-entry [20,22]. To date, there is strong evidence to support the predisposing dyskalemic effects on fatal cardiac arrhythmias and sudden death.

There were some limitations to our current study. Firstly, this was a single-centered retrospective cohort, and the majority of our included patients were Caucasian. Generalization of these findings should be applied with caution. Secondly, a patient’s vital status was obtained from our institutional registry and social security death index database. Thus, data on cause of death at one-year were limited. Future studies are required to identify the cause of death and to identify the preventive measure and follow-up plans after hospitalization to improve one-year survival among those with admission hypo- and hyperkalemia. In addition, the association between admission serum potassium and other clinical outcomes, especially cardiovascular outcomes, should be investigated in future studies.

In conclusion, we demonstrated that serum potassium <4.0 mEq/L or serum potassium >5.0 mEq/L were associated with increased risk of death within 1 year of hospital admission regardless of CKD and CVD.

## Figures and Tables

**Figure 1 medicines-07-00002-f001:**
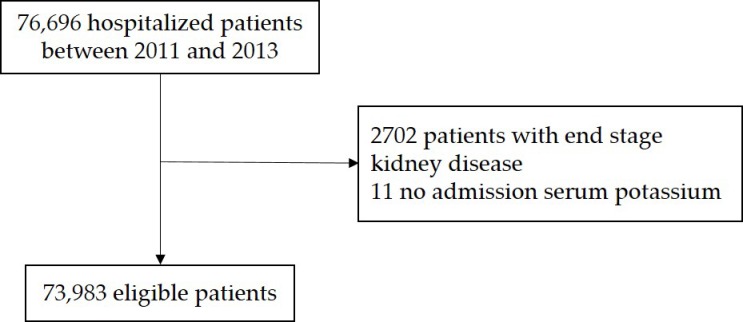
Flow chart of the study.

**Figure 2 medicines-07-00002-f002:**
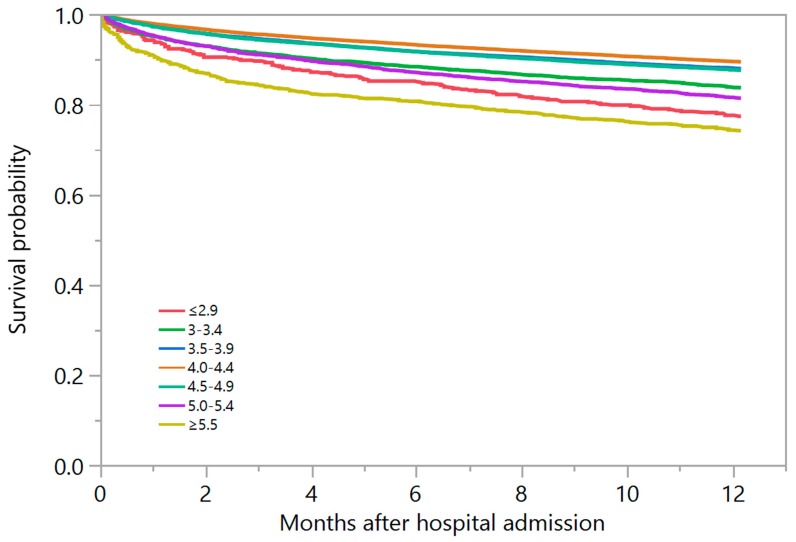
Kaplan–Meier plot of estimated 1-year mortality among patients with different admission serum potassium levels.

**Table 1 medicines-07-00002-t001:** Baseline clinical characteristics.

Variables	All	Serum Potassium Level at Hospital Admission (mEq/L)							
		≤2.9	3.0–3.4	3.5–3.9	4.0–4.4	4.5–4.9	5.0–5.4	≥5.5	*p*
N	73,983	700	3943	17,315	29,815	16,381	4463	1366	
Age (year)	61 ± 18	59 ± 18	60 ± 19	58 ± 19	61 ± 18	63 ± 17	66 ± 16	67 ± 16	<0.001
Male	38,973 (53)	273 (39)	1556 (39)	7999 (46)	16,131 (54)	9522 (58)	2670 (60)	822 (60)	<0.001
Caucasian	68,809 (93)	642 (92)	3562 (9)	15,819 (91)	27,791 (93)	15,497 (95)	4220 (95)	1278 (94)	<0.001
eGFR (mL/min/1.73 m^2^)	80 ± 26	82 ± 34	84 ± 28	86 ± 26	82 ± 25	76 ± 25	67 ± 27	54 ± 27	<0.001
Principal diagnosis									<0.001
- Cardiovascular	15,716 (21)	128 (18)	844 (21)	3747 (22)	6291 (21)	3380 (21)	1019 (23)	307 (22)	
- Endocrine/Metabolic	1879 (3)	75 (11)	137 (3)	384 (2)	558 (2)	459 (3)	158 (4)	108 (8)	
- Gastrointestinal	6920 (9)	80 (11)	537 (14)	1824 (11)	2594 (9)	1335 (8)	429 (10)	121 (9)	
- Hematology/Oncology	11,558 (16)	59 (8)	347 (9)	2000 (12)	4756 (16)	3209 (20)	953 (21)	234 (17)	
- Infectious Disease	2171 (3)	51 (7)	269 (7)	600 (3)	705 (2)	374 (2)	111 (2)	61 (4)	
- Respiratory	2968 (4)	25 (4)	213 (5)	638 (4)	1098 (4)	673 (4)	222 (5)	99 (7)	
- Injury/Poisoning	11,434 (15)	140 (20)	699 (18)	3070 (18)	4573 (15)	2248 (14)	528 (11)	176 (13)	
- Other	21,337 (29)	142 (20)	897 (23)	5052 (29)	9240 (31)	4703 (29)	1043 (23)	260 (19)	
Charlson score	1.7 ± 2.3	1.8 ± 2.5	1.7 ± 2.3	1.5 ± 2.2	1.6 ± 2.2	1.9 ± 2.4	2.3 ± 2.6	2.8 ± 2.7	<0.001
Comorbidities									
- CAD	5260 (7)	38 (5)	227 (6)	1009 (6)	2042 (7)	1347 (8)	423 (9)	174 (13)	<0.001
- CHF	4679 (6)	60 (9)	275 (7)	965 (6)	1665 (6)	1112 (7)	421 (9)	181 (13)	<0.001
- PVD	2065 (3)	20 (3)	106 (3)	371 (2)	736 (2)	552 (3)	199 (4)	81 (6)	<0.001
- Stroke	5228 (7)	52 (7)	315 (8)	1118 (6)	1944 (7)	1247 (8)	398 (9)	154 (11)	<0.001
- DM	13,752 (19)	114 (16)	669 (17)	2743 (16)	5096 (17)	3414 (21)	1206 (27)	510 (37)	<0.001
- COPD	6131 (8)	56 (8)	326 (8)	1197 (7)	2227 (7)	1518 (9)	586 (13)	221 (16)	<0.001
- Cirrhosis	1713 (3)	24 (3)	114 (3)	408 (2)	581 (2)	369 (2)	150 (3)	67 (5)	<0.001
Medications									
- ACEI/ARB	20,068 (27)	162 (23)	974 (25)	4067 (23)	7930 (27)	4745 (29)	1627 (36)	563 (41)	<0.001
- Diuretics	22,020 (30)	341 (49)	1659 (42)	5452 (31)	7974 (27)	4508 (28)	1490 (33)	596 (44)	<0.001
- Potassium supplement	8317 (11)	401 (57)	1025 (26)	2246 (13)	2731 (9)	1336 (8)	431 (10)	147 (11)	<0.001
Acute kidney injury	7602 (10)	113 (16)	404 (10)	1295 (7)	2303 (8)	1955 (12)	953 (21)	579 (42)	<0.001
Mechanical ventilation	6949 (9)	131 (19)	646 (16)	2008 (12)	2498 (8)	1165 (7)	338 (8)	163 (12)	<0.001
Electrolyte at admission									
- Sodium	138 ± 4	137 ± 7	138 ± 5	138 ± 4	138 ± 4	138 ± 4	137 ± 4	136 ± 5	<0.001
- Chloride	103 ± 5	101 ± 11	103 ± 6	103 ± 5	103 ± 4	103 ± 4	103 ± 5	102 ± 6	<0.001
- Bicarbonate	25 ± 3	25 ± 7	25 ± 4	25 ± 3	25 ± 3	25 ± 3	25 ± 4	24 ± 5	<0.001

Continuous data are presented as mean ± SD; categorical data are presented as count (%); ACEI/ARB, angiotensin-converting enzyme inhibitor/angiotensin receptor blocker; CAD, coronary artery disease; CHF, congestive heart failure; COPD, chronic obstructive pulmonary disease; DM, diabetes mellitus; eGFR, estimated glomerular filtration rate; PVD, peripheral vascular disease.

**Table 2 medicines-07-00002-t002:** The association between admission serum potassium and 1-year mortality.

Serum Potassium Level at Admission (mEq/L)	1-Year Mortality (%)	Univariate Analysis		Multivariate Analysis	
		HR (95% CI)	*p*	Adjusted HR (95% CI)	*p*
≤2.9	22.6%	2.34 (1.96–2.79)	<0.001	1.67 (1.39–2.01)	<0.001
3.0–3.4	16.4%	1.66 (1.51–1.82)	<0.001	1.36 (1.23–1.49)	<0.001
3.5–3.9	12.1%	1.16 (1.09–1.24)	<0.001	1.19 (1.11–1.26)	<0.001
4.0–4.4	10.6%	1 (reference)	−	1 (reference)	−
4.5–4.9	12.4%	1.20 (1.12–1.27)	<0.001	1.02 (0.96–1.09)	0.46
5.0–5.4	18.6%	1.87 (1.72–2.04)	<0.001	1.30 (1.19–1.41)	<0.001
≥5.5	25.8%	2.84 (2.52–3.20)	<0.001	1.62 (1.43–1.83)	<0.001

Adjusted for age, sex, race, GFR, principal diagnosis, Charlson comorbidity score, CAD, CHF, PVD, stroke, DM, COPD, cirrhosis, acute kidney injury, mechanical ventilation at admission, admission sodium, chloride, and bicarbonate; HR, hazard ratio.

**Table 3 medicines-07-00002-t003:** Subgroup analysis: Patients admitted with principal diagnosis of cardiovascular disease (*n* = 15,716).

Serum Potassium Level at Admission (mEq/L)	1-Year Mortality (%)	Univariate Analysis		Multivariate Analysis	
		HR (95% CI)	*p*	Adjusted HR (95% CI)	*p*
≤2.9	27.7%	2.94 (2.02–4.27)	<0.001	2.30 (1.57–3.38)	<0.001
3.0–3.4	17.5%	1.77 (1.45–2.17)	<0.001	1.64 (1.34–2.02)	<0.001
3.5–3.9	13.1%	1.21 (1.06–1.38)	0.005	1.29 (1.13–1.47)	<0.001
4.0–4.4	11.0%	1 (reference)	−	1 (reference)	−
4.5–4.9	13.5%	1.25 (1.09–1.43)	0.001	1.02 (0.89–1.16)	0.81
5.0–5.4	18.3%	1.75 (1.46–2.10)	<0.001	1.14 (0.94–1.37)	0.17
≥5.5	26.5%	2.76 (2.15–3.54)	<0.001	1.47 (1.14–1.89)	0.003

Adjusted for age, sex, race, GFR, Charlson comorbidity score, CAD, CHF, PVD, stroke, DM, COPD, cirrhosis, acute kidney injury, mechanical ventilation at admission, admission sodium, chloride, and bicarbonate; HR, hazard ratio.

**Table 4 medicines-07-00002-t004:** Subgroup analysis: Patients with GFR of <60 mL/min/1.73 m^2^ (*n* = 16,710).

Serum Potassium Level at Admission (mEq/L)	1-Year Mortality (%)	Univariate Analysis		Multivariate Analysis	
		HR (95% CI)	*p*	Adjusted HR (95% CI)	*p*
≤2.9	32.5%	1.82 (1.38–2.39)	<0.001	1.94 (1.47–2.56)	<0.001
3.0–3.4	23.6%	1.28 (1.09–1.51)	0.002	1.23 (1.05–1.45)	0.01
3.5–3.9	22.5%	1.18 (1.06–1.31)	0.002	1.22 (1.10–1.36)	<0.001
4.0–4.4	19.6%	1 (reference)	−	1 (reference)	−
4.5–4.9	18.9%	0.97 (0.88–1.06)	0.48	0.93 (0.84–1.02)	0.12
5.0–5.4	25.0%	1.33 (1.18–1.50)	<0.001	1.14 (1.01–1.28)	0.03
≥5.5	29.6%	1.70 (1.48–1.97)	<0.001	1.32 (1.14–1.53)	<0.001

Adjusted for age, sex, race, GFR, principal diagnosis, Charlson comorbidity score, CAD, CHF, PVD, stroke, DM, COPD, cirrhosis, acute kidney injury, mechanical ventilation at admission, admission sodium, chloride, and bicarbonate; HR, hazard ratio.

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
