# Peer review of "Admission Serum Potassium Levels in Hospitalized Patients and One-Year Mortality"

_medicines, 2019, doi:10.3390/medicines7010002_

Round 1

Reviewer 1 Report

Reviewer Comments

In this manuscript the authors assess the relationship between admission serum potassium and long-term mortality in all adult hospitalized patients. The study included 73,983 hospitalized patients primarily admitted due to cardiovascular outcomes. The authors have done an interesting study showing that hypokalemia ≤3.9 mEq/L and hyperkalemia ≥5.0 mEq/L at the time of hospital admission were associated with higher long-term mortality. The manuscript is well-written and the study has been designed properly but there is some missing information that must be addressed.  So, the authors should consider the following comments for a minor improvement prior to re-submission. 

A flow-chart or outline of the study can be included in the method sections to give a clear picture. Although the authors have mentioned the no. of patients admitted in the results section, they should also include it in methods section. How was serum potassium measured in patients? Include in methods section. A total of 73,983 hospitalized patients were included in this study. The mean age was 61±18 years, and 53% were male. The mean eGFR was 80±26 ml/min/1.73 m2. 15,716 (21%) patients were primarily admitted for cardiovascular disease. Did authors find any correlation between serum potassium levels and cardiovascular outcomes in terms of long-term mortality? The patients had cardiovascular disease. What was the baseline blood pressure in all the patients and does it have any correlation with admission serum potassium levels?

Author Response

Response to Reviewer#1

In this manuscript the authors assess the relationship between admission serum potassium and long-term mortality in all adult hospitalized patients. The study included 73,983 hospitalized patients primarily admitted due to cardiovascular outcomes. The authors have done an interesting study showing that hypokalemia ≤3.9 mEq/L and hyperkalemia ≥5.0 mEq/L at the time of hospital admission were associated with higher long-term mortality. The manuscript is well-written and the study has been designed properly but there is some missing information that must be addressed.  So, the authors should consider the following comments for a minor improvement prior to re-submission.

Response: We thank you for reviewing our manuscript and for your critical evaluation.

Comment #1

A flow-chart or outline of the study can be included in the method sections to give a clear picture. Although the authors have mentioned the no. of patients admitted in the results section, they should also include it in methods section.

Response: We agree with the reviewer. Figure 1 has been added to demonstrate the flow-chart of the study as the reviewer's suggestion.

Comment #2

How was serum potassium measured in patients? Include in methods section

Response: We agree with the reviewer. The following statement has been added to describe serum potassium measurement in method section.

Serum potassium was measured using ion-selective electrode (indirect potentiometry) method.  

Comment #3

A total of 73,983 hospitalized patients were included in this study. The mean age was 61±18 years, and 53% were male. The mean eGFR was 80±26 ml/min/1.73 m2. 15,716 (21%) patients were primarily admitted for cardiovascular disease. Did authors find any correlation between serum potassium levels and cardiovascular outcomes in terms of long-term mortality? The patients had cardiovascular disease.

Response: Thank you for your thoughtful suggestion. The data from this study was abstracted from institutional electronic database. Unfortunately, cardiovascular outcomes were not available in our database, and therefore, the association between serum potassium and cardiovascular outcomes was not assessed and reported in this study. This limitation was added in discussion section.

"In addition, the association between admission serum potassium and other clinical outcomes, especially cardiovascular outcomes, should be investigated in future studies."

Comment #4

What was the baseline blood pressure in all the patients and does it have any correlation with admission serum potassium levels?

Response: Thank you for your thoughtful suggestion. Unfortunately, our database did not contain baseline blood pressure. Therefore, we cannot report baseline blood pressure and cannot assess the association between blood pressure and admission serum potassium.

We greatly appreciated the reviewer’s time and comments to improve our manuscript.

Reviewer 2 Report

In this study, authors investigated the association of Admission Serum Potassium Levels in Hospitalized patients and Long-term Mortality and authors found higher mortality associated with serum potassium levels less than 4 and more than 5 mEq/L. I have some major concern with the study. 

As authors pointed out, similar study has been carried out by Collins et.al. from the same institution, and their follow-up was even longer than present study. Therefore, it is hard to find novelty in this article. And long-term is a relative term, and authors followed-up for only 1 year and title is misleading when authors used long-term. 

Author Response

Response to Reviewer#2

In this study, authors investigated the association of Admission Serum Potassium Levels in Hospitalized patients and Long-term Mortality and authors found higher mortality associated with serum potassium levels less than 4 and more than 5 mEq/L. I have some major concern with the study.

Response: We thank you for reviewing our manuscript and for your critical evaluation.

Comment #1

As authors pointed out, similar study has been carried out by Collins et.al. from the same institution, and their follow-up was even longer than present study. Therefore, it is hard to find novelty in this article.

Response: We apologize for the confusion. The study by Collin et al was not conducted in our institution. In addition, our study and Collin et al investigated the association between serum potassium and mortality in different population. Collin et al assessed serum potassium measurement mostly in outpatient setting, whereas our study assessed admission serum potassium in hospitalized patients. Because of acute illness, serum potassium at the time of admission is likely to be deranged, particularly before correction.

Comment #2

And long-term is a relative term, and authors followed-up for only 1 year and title is misleading when authors used long-term.

Response: We agree with the reviewer that long-term is considered relative. We changed long-term mortality to one-year mortality in tile and throughout manuscript to avoid this misleading. We appreciated the reviewer’s important comment.

We greatly appreciated the reviewer’s time and comments to improve our manuscript.

Reviewer 3 Report

Dear Editors of Medicines

I have carefully reviewed the manuscript entitled “Admission Serum Potassium Levels in Hospitalized patients and Long-term Mortality” (medicines-663365) by Charat Thongprayoon.

This single-center retrospective study, which enrolled 73,983 adult hospitalized patients (mean age, 61±18 years, male 53%) with an admission serum potassium level between the years 2011 and 2013, was conducted to assess the relationship between admission serum potassium and long-term mortality in these hospitalized patients.

By using the multivariable Cox proportional hazard analysis, the authors found that both a low admission serum potassium (≤3.9 mEq/L) and elevated admission potassium (≥5.0 mEq/L) were significantly associated with an increased risk of one-year mortality after hospital admission when compared with an admission serum potassium of 4.0-4.4 mEq/L. The results persisted in the subgroup analysis of chronic kidney disease and cardiovascular disease patients.

Generally speaking, this is a well-written manuscript.

The topic of the current study is interesting and of clinical relevance.

The background of the study, study design, data presentation, and interpretation, as well as English writing, are all good.

However, two minor concerns need to be addressed.

#1. What is the 1-year mortality in the current study? I am confused by the two different information (12.6% in “Abstract, line 24, pp.1,” versus 10.1% in “3.3. Admission serum, line 110, pp.6.”

#2. The “911.698” patients should be corrected into “911,698 patients.” (4. Discussion, line149, pp.8)

Author Response

Response to Reviewer#3

This single-center retrospective study, which enrolled 73,983 adult hospitalized patients (mean age, 61±18 years, male 53%) with an admission serum potassium level between the years 2011 and 2013, was conducted to assess the relationship between admission serum potassium and long-term mortality in these hospitalized patients.By using the multivariable Cox proportional hazard analysis, the authors found that both a low admission serum potassium (≤3.9 mEq/L) and elevated admission potassium (≥5.0 mEq/L) were significantly associated with an increased risk of one-year mortality after hospital admission when compared with an admission serum potassium of 4.0-4.4 mEq/L. The results persisted in the subgroup analysis of chronic kidney disease and cardiovascular disease patients.

Generally speaking, this is a well-written manuscript. The topic of the current study is interesting and of clinical relevance. The background of the study, study design, data presentation, and interpretation, as well as English writing, are all good.  

However, two minor concerns need to be addressed.

Response: We thank you for reviewing our manuscript and for your critical evaluation.

Comment #1

What is the 1-year mortality in the current study? I am confused by the two different information (12.6% in “Abstract, line 24, pp.1,” versus 10.1% in “3.3. Admission serum, line 110, pp.6.”

Response: We apologized for the confusion. Estimated one-year mortality was 12.6% based on Kaplan-Meier plot, as stated in the abstract, whereas 7491 (10.1%) out of 73983 died within 1 year, as stated in the result. The number in the result section was revised for consistency to avoid the confusion.

Of 73,983 patients, 12.6% died within one year after hospital admission.

Comment #2

The “911.698” patients should be corrected into “911,698 patients.” (4. Discussion, line149, pp.8)

Response: We appreciated the reviewer’s thorough comment. This error has been corrected.

We greatly appreciated the reviewer’s time and comments to improve our manuscript.

Round 2

Reviewer 2 Report

I have no further comments